# Feasibility, safety, and efficacy of intraoperative magnetic resonance imaging-guided hepatectomy for small hepatocellular carcinoma: A retrospective study

Keiso Matsubara[1], Shintaro Kuroda[1]*, Tsuyoshi Kobayashi[1], Kentaro Ide[1°], Hiroyuki Tahara[1°], Masahiro Ohira[1°], Naruhiko Honmyo[1°], Yuji Akiyama[2], Masataka Tsuge[3], Kazuo Awai[4], Hideki Ohdan[1]

1 Department of Gastroenterological and Transplant Surgery, Graduate School of Biomedical and Health Science, Hiroshima University, Hiroshima, Japan, 2 Department of Radiology, Hiroshima University, Hiroshima, Japan, 3 Department of Gastroenterology and Metabolism, Hiroshima University, Hiroshima, Japan, 4 Department of Diagnostic Radiology, Hiroshima University, Hiroshima, Japan

° These authors contributed equally to this work.

* shintarokuroda@hiroshima-u.ac.jp

**Data Availability Statement:** All relevant data are within the paper.

## Abstract

Advancements in diagnostic modalities, such as enhanced magnetic resonance imaging, provide increased opportunities for identifying small hepatocellular carcinoma that is undetectable on preoperative ultrasonography. Whether it is acceptable to leave these lesions untreated is uncertain. This study aimed to evaluate the safety and efficacy of intraoperative magnetic resonance imaging-guided hepatectomy using new navigation systems. This study was conducted between July 2019 and January 2023. We retrospectively studied the clinicopathological features and prognoses of patients with small hepatocellular carcinoma who underwent curative intraoperative magnetic resonance imaging-guided hepatectomy. We evaluated 23 patients (median age, 75 years), among whom 20 (87.0%) were males. Seven (30.4%) and 15 (65.2%) patients had liver cirrhosis and a history of hepatectomy, respectively. The median size of the target lesions was 9 mm, with a median distance of 6 mm from the liver surface. Despite being undetectable preoperatively on contrast-enhanced ultrasonography, all lesions were identified using intraoperative magnetic resonance imaging. Based on pathological findings, 76.0% of the lesions were malignant. The complete resection rate was 100%, and tumor-free margins were confirmed in 96.0% of the patients. Intraoperative magnetic resonance imaging-guided hepatectomy is safe and effective in identifying and resecting small hepatocellular carcinoma lesions that are undetectable on preoperative ultrasonography.

## Introduction

Hepatocellular carcinoma (HCC) accounts for 70–90% of primary liver cancers and is the third leading cause of cancer-related death [1–3]. Very early HCC is a single tumor <2 cm in diameter in a patient with asymptomatic Child–Pugh class A liver disease [4].

**Funding:** The author(s) received no specific funding for this work.

**Competing interests:** The authors have declared that no competing interests exist.

**Abbreviations:** HCC, hepatocellular carcinoma; AASLD, the American Association for the Study of Liver Diseases; US, ultrasonography; CEUS, contrast-enhanced ultrasonography; EOB-MRI, gadolinium-ethoxybenzyl-diethylenetriamine pentaacetic acid-enhanced magnetic resonance imaging; IOUS, intraoperative ultrasonography; iMRI, intraoperative magnetic resonance imaging; MRI, magnetic resonance imaging; CECT, contrast-enhanced computed tomography; CTHA, computed tomography during hepatic arteriography; CTAP, computed tomography during arterial portography; SCOT, Smart Cyber Operating Theater; CEIOUS, contrast-enhanced intraoperative ultrasonography; DNs, dysplastic nodules; FNH, focal nodular hyperplasia; RVS, real-time virtual sonography.

Although the American Association for the Study of Liver Diseases (AASLD) recommends ultrasonography (US) for initial HCC screening for suspicious nodules, it is frequently difficult to detect very early HCC using US [5]. Advancements in diagnostic modalities, including Sonazoid contrast-enhanced US (CEUS) and gadolinium-ethoxybenzyl-diethylenetriamine pentaacetic acid-enhanced magnetic resonance imaging (EOB-MRI), have provided higher degrees of detectability for small HCC undetectable with US [6].

However, the optimal therapeutic strategy for very early-stage HCC remains to be determined. Although the current AASLD guidelines recommend liver surgical resection or ablation as the main treatment for patients with Child–Pugh A and small HCC [5], the intraoperative diagnosis of small lesions remains insufficient [7, 8]. For these small lesions, performing a partial hepatectomy without anatomical landmarks makes intraoperative tumor identification extremely challenging, and there is always the possibility of a wider extent of hepatectomy. In some cases, the lesion may remain unresected due to undetectability with intraoperative US (IOUS). If the lesion cannot be completely resected during the surgery, it may lead to early recurrence and significant disadvantages. Thus, novel surgical technologies and medical devices are required to resect these tumors.

Since its introduction in mid-1990s, intraoperative magnetic resonance imaging (iMRI) has undergone several enhancements over the last several decades and has been particularly beneficial to the practice of neurosurgical oncology [9, 10]. An open configuration magnetic resonance imaging (MRI) scanner of "double-doughnut" magnets allows surgery to be performed with concurrent intraoperative image [11]. iMRI identifies the lesion and assesses the degree of surgical achievement; to date, >60 iMRI systems have been installed worldwide, with thousands of surgeries performed. These techniques can provide the highest quality evaluation of surgical execution and assessment of the dynamic changes that occur during surgery in near real-time. iMRI-guided hepatectomy offers the potential to provide a cure for patients with small HCC that is only recognizable through MRI preoperatively. Therefore, we investigated whether this procedure can be applied to hepatectomies for small liver tumors. This study aimed to evaluate the effectiveness and safety of iMRI-guided hepatectomy with new navigation systems for these small liver tumors. To the best of our knowledge, this is the first series of reports on iMRI-guided hepatectomy for small HCC.

## Materials and methods

This exploratory study included patients who underwent iMRI-guided hepatectomy at Hiroshima University Hospital in Hiroshima, Japan, between July 2019 and January 2023. This study was conducted in accordance with the principles of the Declaration of Helsinki. This study and its associated protocol were not registered with the University Hospital Medical Information Network or Clinical Trials.gov, although they received approval from the ethics committee of the institution (approval number: E-1600, Hiroshima University). All data were fully anonymized before we analyzed them. Patients' medical records were retrospectively reviewed, and data were collected. Patients provided written consent after being informed of the purpose and investigational nature of this study. We retrospectively collected the demographic and clinicopathological data of consecutive patients, including physical status, age, sex, tumor marker levels, operation time, surgical duration, blood loss, liver tumor pathology, tumor-free margins, length of hospitalization, postoperative complications, and complete resection rate. The surgical margin was assessed based on the distance from the lesion closest to the cut surface of the liver, and it was macroscopically classified as tumor-free margin if the distance was ≥0 mm and positive margin if the tumor was clearly exposed on the cut surface.

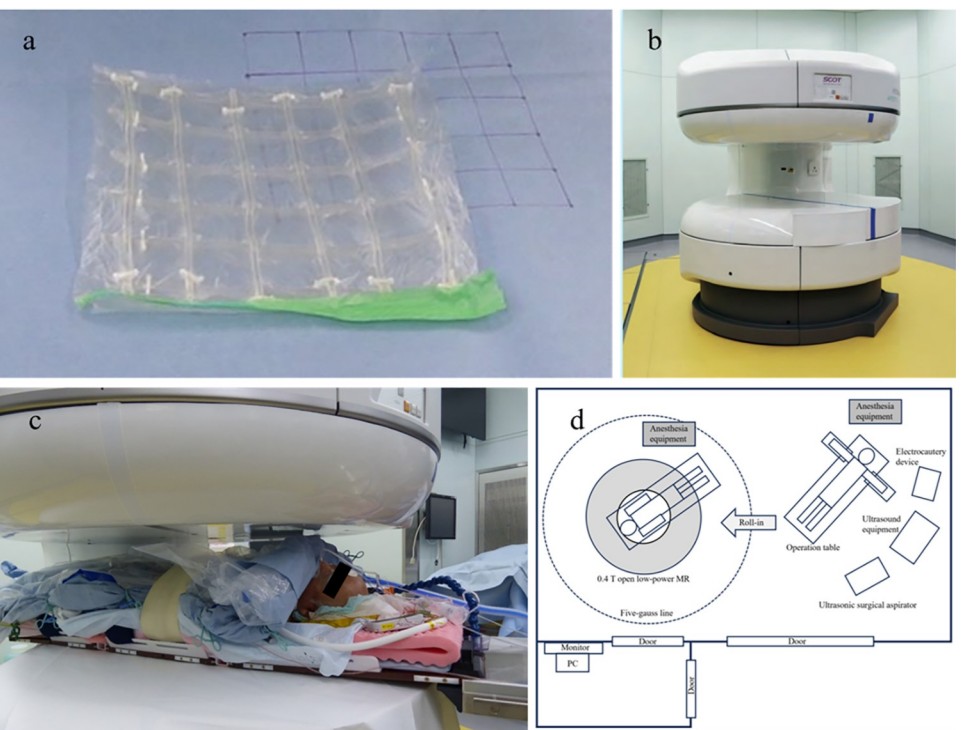

**Fig 1. Grid marker and intraoperative magnetic resonance imaging (iMRI).** (a) Grid marker for iMRI comprising 12 polyvinyl chloride Nelaton catheters (approximately 10 cm in length and 10 Fr in outer diameter each). (b) iMRI using a 0.4 T open low-power MR (APERTO Lucent, FUJIFILM Healthcare Corporation, Tokyo, Japan). (c) Intraoperative view of the setting for iMRI during hepatectomy. (d) Scheme of the Smart Cyber Operating Theater. The five-gauss line refers to an area or boundary around an electromagnetic source where the strength of the magnetic field drops to ≤5 gausses.

## Grid marker for iMRI

This iMRI marker can be used as a positional index for MRI. Twelve polyvinyl chloride Nela-ton catheters (approximately 10 cm long, 10 Fr in outer diameter) were arranged in a grid pattern on a plane with an interval of 2 cm, and their positions were fixed to each other using a nonmagnetic synthetic resin sheet (Fig 1A). The tubes were filled with an MRI contrast agent (gadoterate meglumine 38% aqueous solution, diluted 100 times with physiological saline), which could be detected as signals on MRI.

## Indication for hepatectomy

Preoperative investigations included contrast-enhanced computed tomography (CECT), CT during hepatic arteriography (CTHA), CT during arterial portography (CTAP), EOB-MRI, Sonazoid CEUS, and blood tests. Although EOB-MRI is highly sensitive, it can produce false positives. Therefore, a multimodal approach, frequently combining EOB-MRI with CECT or CTAP/CTHA, is used to identify hypervascular nodules designated for surgical resection. Following consultations with hepatologists, particular attention was provided to lesions suspected to be HCC with a diameter of <2 cm that may not be conclusively identified via preoperative Sonazoid CEUS. In such cases, iMRI-guided hepatectomy was considered. For hypovascular tumors, in addition to the criteria mentioned, those displaying growth tendencies of >15 mm in size were also evaluated as candidates for resection due to their heightened risk of vascularization. Patients underwent hepatectomy, and the choice of resection was based on tumor size

and location and liver function. The surgeries were performed by a board-certified expert surgeon specialized in the hepatobiliary-pancreatic field in Japan [12].

## Surgical procedure

We performed iMRI-guided hepatectomy in the Smart Cyber Operating Theater (SCOT) [13, 14] operating system with 0.4 T open low-power MR (APERTO Lucent, FUJIFILM Healthcare Corporation, Tokyo, Japan) (Fig 1B–1D). Surgery was performed through an upward midline incision, and a subcostal incision was also made when necessary. Sonazoid contrast-enhanced IOUS (CEIOUS) was routinely performed to confirm the presence of all tumors (Fig 2A). To identify the tumors, we performed the first intraoperative EOB-MRI after a new patented grid marker for MRI was placed on the liver surface based on preoperative MRI information, and four points were fixed on the liver surface with 5–0 Prolene (Fig 2B–2D). Hepatic parenchymal transection was performed using an ultrasonic surgical aspirator and bipolar forceps coagulation with the Pringle maneuver based on the grid marker information from the initial iMRI (Fig 2E and 2F). After the hepatectomy, we performed a second iMRI to confirm the resection of the liver tumor (Fig 2G). In cases where iMRI after hepatectomy was impossible because of facility reasons, a second MRI was performed on postoperative day 1 to confirm complete resection. If residual tumors were found after the initial liver resection, additional iMRI-guided hepatectomy was performed. In cases of multiple tumors with lesions that were resectable through simultaneous systematic resection and iMRI-guided hepatectomy, we performed iMRI-guided hepatectomy in addition to the usual hepatectomy.

## Data collection

The evaluation items included the following: complete resection rate (intention-to-treat); surgical results, including tumor identification rate of IOUS and iMRI; operation time; imaging

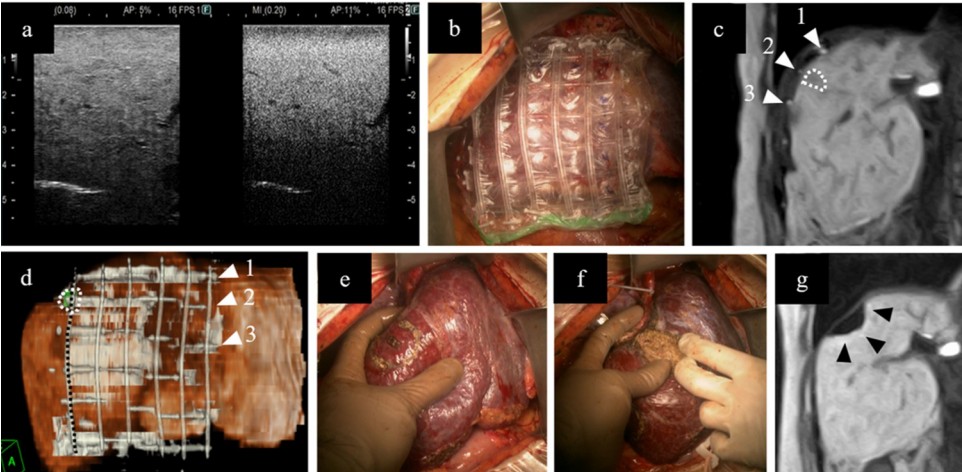

**Fig 2. Intraoperative magnetic resonance imaging (iMRI)-guided hepatectomy.** (a) Sonazoid contrast-enhanced intraoperative ultrasound sonography for the lesion that could not be identified during the surgery. (b) Grid marker placed corresponding to the tumor location based on the preoperative imaging diagnosis. (c) Initial gadolinium-ethoxybenzyl-diethylenetriamine pentaacetic acid-enhanced iMRI (0.4T) performed to confirm the relationship between the grid marker (white triangle) and tumor (white dotted line). All target lesions were identified as tumors showing hypointensity in the hepatobiliary phase. All non-target lesions confirmed on preoperative MRI were also confirmed using iMRI. (d) Three-dimensional image constructed based on the initial iMRI, enabling a visual understanding of the tumor location. The tumor (white dotted line) is located on the second grid from the top (white triangle) on the left end (black dotted line) of the lattice-like structure. (e) Planned resection margin based on the initial iMRI. (f) Results following iMRI-guided hepatectomy. (g) The second iMRI performed after hepatectomy, confirming successful tumor resection (black triangle).

time; bleeding amount; marker visibility; tumor identification rate after marking; complications; and concordance rate of preoperative and pathological diagnoses. Complete resection was defined as the absence of gross or radiographic tumor remnants after hepatectomy. Postoperative complications were defined as complications occurring within 30 days postoperatively and graded according to the Clavien–Dindo classification; those with a Clavien–Dindo grade of ≥III were defined as major complications [15].

## Results

During the study period from July 2019 to January 2022, 23 patients underwent iMRI-guided hepatectomy for 25 liver tumors measuring <20 mm, which were preoperatively identified using MRI and other modalities excluding US. The demographic characteristics of the study population are presented in Table 1. The median age of the patients was 75 (range, 60–87) years, and 20 (87.0%) were males. Among these patients, 11 (47.8%) were positive for hepatitis C virus antibody, 4 (17.3%) were positive for hepatitis B surface antigen, and 7 (30.4%) had cirrhosis. Twenty-two (95.6%) and 15 (65.2%) patients had Child–Pugh grade A disease and a surgical history of hepatectomy, respectively. The median number of liver tumors was 2 (range, 1–4).

Among all the patients, 22 (95.7%) underwent iMRI-guided partial hepatectomy and 1 (4.3%) underwent iMRI-guided subsegmentectomy for the target lesions. Furthermore, in 16

**Table 1. Patients' characteristics.**

|  | N = 23 |
|---|---|
| Age, years* | 75 (60–87) |
| Sex, male/female | 20/3 |
| Background liver, NL/CH/LC | 6/10/7 |
| Etiology, HBV/HCV/NBNC | 4/11/8 |
| History of hepatectomy, |  |
| no/yes | 8/15 |
| Child–Pugh, A/B | 22/1 |
| Number of tumors* | 2 (1–4) |
| Number of target tumors* | 1 (1–2) |
| Operation time, min* | 468 (311–683) |
| Blood loss, mL* | 705 (70–1930) |
| Surgical procedure, n (%) |  |
| Target lesions |  |
| Partial hepatectomy | 22 (95.7) |
| Subsegmentectomy | 1 (4.3) |
| Non-target lesions |  |
| None | 7 (30.4) |
| Partial hepatectomy | 10 (43.5) |
| Subsegmentectomy | 4 (17.4) |
| Sectionectomy | 2 (8.7) |
| Major complications, N (%) | 1 (4.3) |
| Postoperative hospital stay, days* | 11 (9–34) |

*Data are presented as medians (ranges).

Abbreviations: NL/CH/LC, normal liver/chronic hepatitis/liver cirrhosis; HBV/HCV/NBNC, hepatitis B virus/hepatitis C virus/non-B non-C liver disease

(69.6%) patients, additional hepatectomy was performed in the standard manner for non-target lesions, concurrently with iMRI-guided primary treatment. The surgical procedures for non-target lesions were as follows: 10 (43.5%), 4 (17.4%), and 2 (8.7%) patients had partial hepatectomy, subsegmentectomy, and sectionectomy, respectively. The median operation time and blood loss were 468 (range, 311–683) min and 705 (range, 70–1930) mL, respectively. One (4.3%) patient experienced the major complication of pneumonia, with a mortality rate of 0%. The median duration of postoperative hospital stay was 11 (range, 9–34) days.

The median size of the 25 target lesions was 9 (range, 5–15) mm (Table 2). The median distance from the liver surface to the tumor was 6 (range, 0–19) mm. Nevertheless, all lesions

**Table 2. Target lesions for intraoperative magnetic resonance imaging-guided hepatectomy.**

|  | N = 25 |
| --- | --- |
| Imaging factors |  |
| Target tumor diameter, mm* | 9 (5–15) |
| Location of target tumors |  |
| Segment III | 2 |
| Segment IV | 2 |
| Segment V | 3 |
| Segment VI | 2 |
| Segment VII | 3 |
| Segment VIII | 13 |
| Distance from liver surface to tumors, mm* | 6 (0–19) |
| Distance from liver surface to tumors > 1 cm, n (%) | 17 (68.0) |
| Distance from liver surface to tumors ≤ 1 cm, n (%) | 8 (32.0) |
| Surgical factors |  |
| Surgical procedure, n (%) |  |
| Partial hepatectomy | 24 (96.0) |
| Subsegmentectomy | 1 (4.0) |
| Total number of iMRI imaging* | 2 (1–2) |
| First iMRI imaging duration, min* | 60 (48–80) |
| Second iMRI imaging duration, min* | 43 (36–56) |
| Total iMRI imaging duration, min* | 102 (56–115) |
| Identification rate of CEIOUS | 36.0% (9/25) |
| Identification rate of iMRI | 100% (25/25) |
| Complete resection rate | 100% (25/25) |
| Pathological findings |  |
| Pathological diagnosis |  |
| Adenoma/DN/FNH/HCC | 1/4/1/19 |
| Histological type of HCC |  |
| Well-differentiated type | 10 |
| Moderately differentiated type | 8 |
| Poorly differentiated type | 1 |
| MVI of HCC, n (%) | 3 (12.0%) |
| Tumor-free margin positive/negative/unknown | 0/24/1 |
| Tumor-free margin, mm* | 3.8 (0–17) |

*Data are presented as medians (ranges).

Abbreviations: iMRI, intraoperative magnetic resonance imaging; CEIOUS, contrast-enhanced intraoperative ultrasonography; DN, dysplastic nodule; FNH, focal nodular hyperplasia; HCC, hepatocellular carcinoma; MVI, microvascular invasion

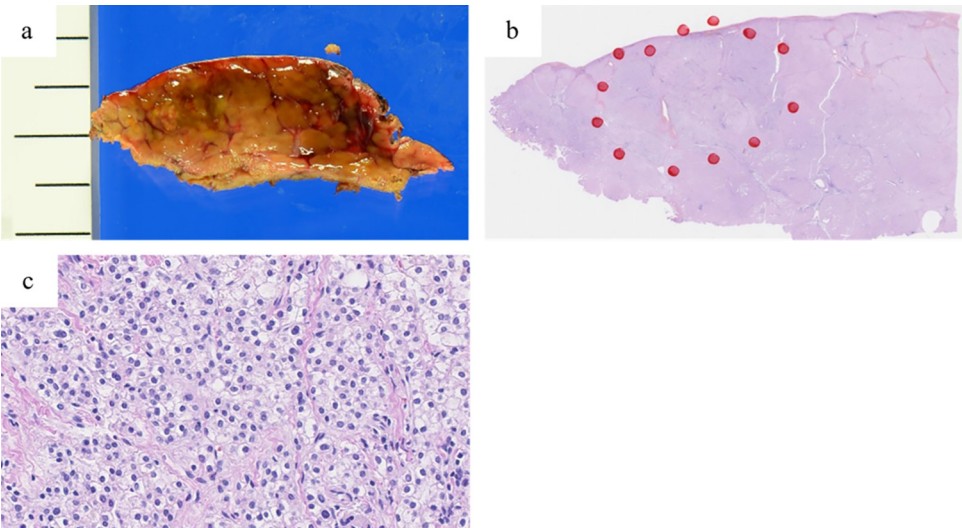

**Fig 3. Pathological findings of the small hepatocellular carcinoma.** (a) Cut surface of the tumor. (b) Specimen in a low-power field (hematoxylin and eosin, original magnification ×40). (c) Specimen in a high-power field comprising well-differentiated hepatocellular carcinoma (hematoxylin and eosin, original magnification ×100).

were undetectable through CEUS preoperatively, nine (39.1%) were detectable with CEIOUS, and all were identified using iMRI. The median total number of iMRI scans was 2 (range, 1–2), and the median durations of the first and second iMRI scans were 60 (range, 48–80) and 43 (range, 32–56) min, respectively. Repeat iMRI was not required because there were no discrepancies between the markers and tumor locations. The complete resection rate was 100%.

## Pathological findings of target lesions

Pathological findings for 25 target lesions revealed that 19 lesions were malignant (HCC) and 6 were benign, including four dysplastic nodules (DNs), one focal nodular hyperplasia (FNH), and one adenoma. The rate of concordance of preoperative diagnosis for HCC was 76.0% (Table 2). Complete resection was accomplished in all patients. Tumor-free margins were confirmed in 24 (96.0%) lesions, and one was difficult to evaluate. Of the 19 lesions pathologically diagnosed as HCC (Fig 3A–3C), 10 (52.6%) were histologically well-differentiated, eight (42.1%) were moderately differentiated, one (5.3%) was poorly differentiated, and three (12.0%) showed microvascular invasion (vp1).

## Discussion

To the best of our knowledge, this is the first report on a series of iMRI-guided hepatectomies with liquid-containing grid markers to investigate the safety and efficacy of iMRI-guided hepatectomy. This report suggests that the described procedure can achieve R0 resection for patients with small liver lesions that should be resected but are undetectable by IOUS.

Improvements in multimodal therapeutic strategies and the development of surgical techniques for treating malignant liver tumors have led to advancements in liver surgery. Recently, EOB-MRI and Sonazoid CEUS have been developed for focal liver lesions, both of which can provide late-phase images for hemodynamic evaluation. The improvement in these two imaging modalities can provide a higher sensitivity for diagnosing nodules of <2 cm. Qin et al. reported that the diagnostic sensitivities of EOB-MRI and CEUS for small HCC (<2 cm) were 61.0% and 41.5%, respectively [6]. MRI with hepatocyte-specific contrast medium can be

effective for differentiating between HCC and regenerative nodules in liver cirrhosis, and EOB-MRI can be useful for small HCC because smaller tumor size causes lower spatial and contrast resolution, and the objectivity of EOB-MRI is higher than that of CEUS [16].

Inspection, palpation, and IOUS are standard methods for the intraoperative localization of liver tumors in open liver surgery. IOUS is superior to preoperative multidetector CT and EOB-MRI for detecting liver tumors [17]; however, we encountered liver tumors that were visible only through MRI rather than via preoperative US. In this study, of the 25 target lesions, none could be confirmed on preoperative CEUS and only 9 (39.1%) were identified on CEIOUS; in contrast, all lesions were detected on iMRI. Conversely, all non-target lesions initially detected through preoperative CEUS were consistently confirmed using iMRI. Although this is a single-arm study, iMRI demonstrated the potential to detect small liver tumors comparably to preoperative CEUS. Moreover, more than half of the target lesions that could not be identified during CEIOUS were successfully visualized using iMRI. The achievement of such a high detection rate with a 0.4T iMRI can be partially attributed to the use of general anesthesia, which enables precise breath control and complete immobilization during imaging, resulting in the acquisition of high-precision images. Regarding intraoperative palpation, all cases in this series underwent open surgery. Importantly, there were no nodules that could be identified through palpation but remained undetectable with CEIOUS.

Recently, other techniques for the intraoperative detection of hepatic lesions have been implemented and should be briefly discussed. Different methods of intraoperative image guidance exist, including indocyanine green (ICG) fluorescence imaging or intraoperative real-time three-dimensional navigation, which can be used to detect intrahepatic lesions [18]. ICG fluorescence imaging of HCC during hepatectomy has improved; however, some limitations exist in detecting small lesions. First, the slower metabolic elimination of ICG can lead to an increased false positive rate in detecting tumors in livers affected by cirrhosis or fibrosis, which have impaired function. Second, it is challenging to detect lesions, particularly those located >5–10 mm distant from the liver surface [18–22]. In this study, seven (30.4%) patients had cirrhosis and eight (32.0%) of the target lesions were located deeper than 1 cm from the liver surface. iMRI-guided hepatectomy may be more effective than hepatectomy with ICG fluorescence imaging in resecting lesions in cirrhotic livers or those situated deeper within the tissue. Tumors located deep within the liver, such as those near the root of the Glisson's capsule, frequently require anatomical resection. Therefore, iMRI-guided hepatectomy appears to be most effective for cases involving tumors in the intermediate area, where partial resection is recommended. Real-time virtual sonography (RVS), developed to enhance the interpretation of IOUS images, provides synchronized two-dimensional CT images alongside real-time US images on a monitor [23–25]. RVS is particularly valuable for visualizing and understanding vascular structures during anatomical liver resections. However, when dealing with liver tumors, such as the target lesions in our study, which may not be readily visible or can pose challenge in identifying using IOUS, determining the transection line for partial resection is difficult. Additionally, the practical application of the RVS system may be limited, as it requires the liver to be repositioned to its original orientation. In deep anatomical locations, especially in the right subphrenic space for examining liver segments VII and VIII, the relatively large attachment of the IOUS probe may impose restrictions on the use of RVS [26]. In such cases, all lesions can be resected using iMRI-guided hepatectomy. Thus, iMRI-guided hepatectomy can be an effective alternative, potentially addressing the limitations of other methods and enabling surgical excision beyond the cirrhotic liver or tumors in deep anatomical locations. Re-performing iMRI before and after hepatectomy was not necessary because of the deviation in the positional relationship between the marker and tumor. Furthermore, tumor localization and resection were possible in all cases; therefore, this new marker can be considered useful.

To the best of our knowledge, this is the first report on a series of iMRI-guided hepatectomies. Riediger et al. reported the first intraoperative use of iMRI for confirming the excision of the liver tumor while associating liver partition and portal vein ligation for staged hepatectomy [27]. iMRI was safe and feasible even in a patient with a provisory-closed abdomen during liver surgery. In our series of iMRI-guided hepatectomies with liquid-containing markers, no marking-related complications occurred during surgery. However, the operation time tended to be lengthy, with an average of 468 min, and there was a tendency for a substantial amount of bleeding. The causes of the prolonged operation time and large amount of bleeding include cases with repeated hepatectomies that require adhesion detachment, cases with multiple nodules, and iMRI imaging time. There were seven (28.0%) cases with bleeding of >1 L, with most of them involving repeated or multiple hepatectomies. The total operation time extended by approximately 100 min compared with the standard hepatectomy due to the inclusion of two iMRI sessions. Regarding postoperative complications, the major complication and morbidity rates were 4.3% and 0%, respectively, which are lower than those reported in previous studies [28]. The length of hospital stay was comparable to that of previous reports [28].

The rate of achieving complete resection with a single iMRI-guided hepatectomy was 87.0%. Moreover, with additional iMRI-guided hepatectomy, complete resection was achieved in all patients. Tumor-free margins were confirmed for all target lesions, except for one that was difficult to evaluate due to heat denaturation during resection. Considering that the median of the tumor-free margin was 3.8 (range, 0–17 mm), an excessive resection was unlikely to have occurred during iMRI-guided hepatectomy. Regarding the pathological examinations for the target lesions, the rate of concordance of the preoperative diagnosis was as high as 76.0%. Because tumors diagnosed as DN are likely to progress to HCC in the future and those diagnosed as FNH are difficult to distinguish from HCC preoperatively, this procedure is considered useful even for the resection of these tumors. Although it may be highlighted that this procedure may not be necessary if systematic resection can be performed, this study included patients with cirrhosis who had lower liver function and could only undergo partial hepatectomy. The prognostic effect of preoperative US on the resection of unidentifiable nodules remains unknown because the prognosis was not directly compared with that of unresectable cases. However, for these small tumors, there is a possibility of excessively wide hepatectomy; moreover, in such cases, the tumor may not be resectable owing to the undetectability of IOUS. This procedure was considered useful because 56.7% of the target lesions were only resectable through this procedure. Among the 23 patients, seven (30.4%) presented only target lesions. Serial monitoring and follow-up with imaging for these tumors to grow until it is detectable through US before proceeding with surgery may be considered acceptable. However, considering reports that the prognosis after radical resection for early-stage HCC (including Barcelona Clinic Liver Cancer Stage 0) is favorable and scoring systems indicating that a tumor diameter of ≥2 cm is a prognostic risk factor [29, 30], performing curative resection for early-stage HCC can be one of the treatment options. Among the 23 patients, 16 (69.7%) patients who had not only target lesions but also other non-target lesions identified as the main tumors detectable through preoperative US achieved R0 resection. Pathological examination showed that 76.0% of the excised target lesions were malignant, suggesting a potential for preventing early recurrence; thus, such patients benefited from this procedure.

iMRI-guided hepatectomy is restricted to facilities that can perform MRI during surgery. Therefore, enhancing the surgical environment is essential for its broader adoption. SCOT serves as a therapeutic interface, connecting different devices in the operating room, including intraoperative imaging devices and surgical instruments, enabling equipment networking [13, 14]. Although surgery was performed using SCOT in a single room, we believe that this

procedure can be performed without relying on a networked system. If iMRI is available, it can be performed successfully. There are facilities equipped with separate rooms specifically designed for iMRI. Moreover, this procedure requires the preparation and management of the operating room environment and MRI-compatible equipment. Additionally, it extends the overall surgical time, requires a larger workforce, and incurs higher operational costs. In this study, the setup and execution of iMRI typically required approximately 60 min each. Therefore, iMRI-guided hepatectomy is unsuitable for urgent surgeries. Adjustments to equipment and costs are necessary; however, with advancements in diagnostic techniques for small HCC, there is potential for further widespread adoption in the future.

This study has some limitations. The retrospective design and the absence of randomization could have limited the power and precision of our results. The prognostic effect of resecting small HCCs that are undetectable through preoperative US on the prognosis is unknown because it cannot be directly compared with cases after the nodules have progressed to be identified. This epidemiological study showed safety and efficacy in 23 patients; therefore, further large-scale trials are necessary to verify the safety and efficacy of this procedure. Furthermore, a prospective clinical trial is required to directly compare tumor detection capabilities between CEIOUS and iMRI. It might have been an excessive application of iMRI for 36.0% of the target lesions that were identifiable through CEIOUS; however, this could not have been predicted preoperatively. Nevertheless, with an increasing number of cases in the future, we may identify trends in tumors that cannot be identified through preoperative US but can be detected with CEIOUS, enabling us to select suitable candidates for iMRI-guided hepatectomy. Although the implementation was performed through open surgery in this study, the application of this modality to minimally invasive surgery, including laparoscopy, is currently under development and is a challenge worth addressing.

## Conclusions

iMRI-guided hepatectomy is beneficial for identifying and resecting tumors suspicious of small HCC that are undetectable during preoperative US.

## Acknowledgments

We thank Editage (www.editage.com) for the English language editing services.

## Author Contributions

**Conceptualization:** Shintaro Kuroda, Tsuyoshi Kobayashi.

**Data curation:** Keiso Matsubara, Shintaro Kuroda.

**Formal analysis:** Keiso Matsubara, Shintaro Kuroda.

**Funding acquisition:** Hideki Ohdan.

**Investigation:** Shintaro Kuroda, Tsuyoshi Kobayashi, Kentaro Ide, Hiroyuki Tahara, Masahiro Ohira, Naruhiko Honmyo.

**Methodology:** Shintaro Kuroda, Tsuyoshi Kobayashi.

**Project administration:** Shintaro Kuroda.

**Resources:** Shintaro Kuroda, Tsuyoshi Kobayashi, Yuji Akiyama, Masataka Tsuge, Kazuo Awai.

**Supervision:** Shintaro Kuroda, Tsuyoshi Kobayashi, Hideki Ohdan.

**Validation:** Shintaro Kuroda.

**Visualization:** Shintaro Kuroda.

**Writing – original draft:** Keiso Matsubara.

**Writing – review & editing:** Shintaro Kuroda, Tsuyoshi Kobayashi, Kentaro Ide, Hiroyuki Tahara, Masahiro Ohira, Naruhiko Honmyo, Yuji Akiyama, Masataka Tsuge, Kazuo Awai, Hideki Ohdan.

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
