## [Decision Letter · Decision Letter 0]

21 May 2024

PONE-D-24-13064Feasibility, safety, and efficacy of intraoperative magnetic resonance imaging-guided hepatectomy for small hepatocellular carcinoma: A retrospective studyPLOS ONE

Dear Dr. Kuroda,

Thank you for submitting your manuscript to PLOS ONE. After careful consideration, we feel that it has merit but does not fully meet PLOS ONE’s publication criteria as it currently stands. Therefore, we invite you to submit a revised version of the manuscript that addresses the points raised during the review process.

We look forward to receiving your revised manuscript.

Kind regards,

Takehiko Hanaki, MD, PhD

Academic Editor

PLOS ONE

Journal Requirements:

2. In the online submission form, you indicated that [All study data are available from the corresponding author upon reasonable request.]. 

4. We note that Figure(s) 1a, 1b, 1c, 2 and 3 in your submission contain copyrighted images. All PLOS content is published under the Creative Commons Attribution License (CC BY 4.0), which means that the manuscript, images, and Supporting Information files will be freely available online, and any third party is permitted to access, download, copy, distribute, and use these materials in any way, even commercially, with proper attribution. For more information, see our copyright guidelines: http://journals.plos.org/plosone/s/licenses-and-copyright.

a. You may seek permission from the original copyright holder of Figure(s) 1a, 1b, 1c, 2 and 3 to publish the content specifically under the CC BY 4.0 license. 

Additional Editor Comments:

Additional comments as Editor in response to reviewer 1's comments.

This paper examines the use of iMRI as a novel diagnostic method for small hepatocellular carcinoma and its effectiveness during open surgery. Although the novelty of the intraoperative diagnostic method is certainly present, there is concern that it is an open approach as a necessary circumstance in the current situation where MIS is widely used. However, the limitations of the indications for minimally invasive surgery have already been discussed in the limitations section.

In addition, cost-effectiveness does not need to be discussed, as it is based on existing MRI and does not significantly undermine the novelty of the present manuscript.

Reviewers' comments:

Reviewer's Responses to Questions

**Comments to the Author**

1. Is the manuscript technically sound, and do the data support the conclusions?

Reviewer #1: Yes

Reviewer #2: Yes

Reviewer #3: Yes

2. Has the statistical analysis been performed appropriately and rigorously? 

Reviewer #1: Yes

Reviewer #2: Yes

Reviewer #3: Yes

3. Have the authors made all data underlying the findings in their manuscript fully available?

Reviewer #1: Yes

Reviewer #2: Yes

Reviewer #3: Yes

4. Is the manuscript presented in an intelligible fashion and written in standard English?

Reviewer #1: Yes

Reviewer #2: Yes

Reviewer #3: Yes

5. Review Comments to the Author

Reviewer #1: The authors present an interesting article utilizing intraoperative MRI guided hepatectomy for small HCCs and had good results doing so.

I have 2 main questions: 1) Most patients with HCC <2cm we would not perform an open hepatectomy on today in the United States. Are the authors arguing the earlier resection of HCC <2cm will lead to better outcomes? If so, what are the survival and morbidity outcomes for these patients? 2) What is the cost effectiveness of utilizing an iMRI vs. serial monitoring and f/u with imaging prior to resection once the HCC reaches a larger size? What are the survival outcomes of such as well? The authors discuss tumor free margins but no long-term outcomes

This study is obviously in comparison to pre-operative ultrasound but what about intra-operative ultrasound? We often confirm all liver lesions with intra-operative ultrasound as it is cheap and widely available at our institution.

If the HCCs are <2cm, why were these performed open vs. minimally invasive?

This project, as is, does not seem to lead to decreased cost or improved outcomes for the patients given the current study.

Reviewer #2: 1. The topic is not unique but worthy of researching

2. There are many papers in google scholar and Refseek about this topic since 2020

3. Ethical approval is mentioned

4. the title is attractive

5. The abstract is informative

6. The aim is clear

7. The KEYWORDS are good

8. Please use http://www.ncbi.nlm.nih.gov/mesh

9. Lack of the abbreviations section

10. The introduction provides sufficient background information for readers in the immediate field to understand the problem/hypotheses

11. The text arrangement is good

12. The method section is good

13. The depth of the academic material is good

14. The study design is good

15. The suitability and accuracy of questions is good

16. The research methodology is good

17. The materials are good

18. The logic is clear

19. The paper is novel

20. There are a few grammatical errors in this article

21. The related concepts are introduced

22. The readability is sufficient

23. The results are good

24. All tables are clear enough to summarize the results for presentation to the readers

25. All tables are well referred to in the text

26. Lack of figures? Please add figures.

27. The theoretical analysis in this article is sufficient

28. The discussion of results from multiple angles is sufficient

29. The conclusion is tenable

30. The reference section contains too many old references

31. Please use (google scholar and Refseek) search engines then set it since 2020

32. The references are in order within the text

33. Bias is present

34. There is no conflict of interest with the author about this topic

35. Fund is mentioned

36. Conflict of interest is mentioned

37. Acknowledgement is mentioned

38. You can use my suggestions

My final decision is acceptable after minor revision

Reviewer #3: The work presented by you is commendable and gives insight into a new modality of treatment for small HCCs of the liver. The manuscript is well written in a logical flow. The results have been written in an organized way discussion is quite relevant. Conclusion is well written.

6. PLOS authors have the option to publish the peer review history of their article (what does this mean?). If published, this will include your full peer review and any attached files.

Reviewer #1: No

Reviewer #2: **Yes: **hazim alhiti

Reviewer #3: No

---

## [Author Response · Author response to Decision Letter 0]

13 Jun 2024

Manuscript. Number: PONE-D-24-13064

Title: Feasibility, safety, and efficacy of intraoperative magnetic resonance imaging-guided hepatectomy for small hepatocellular carcinoma: A retrospective study

Journal: PLOS ONE

Reviewer #1 

Reviewer#1 point #1: 

Most patients with HCC <2cm we would not perform an open hepatectomy on today in the United States. Are the authors arguing the earlier resection of HCC <2cm will lead to better outcomes? If so, what are the survival and morbidity outcomes for these patients?

Author response #1: 

Thank you for your suggestion. We performed iMRI-guided hepatectomy for lesions that were undetectable via preoperative Sonazoid CEUS, which were likely to be unresectable. It is not possible to directly compare the long-term postoperative outcomes of these small lesions with those that have grown sufficiently large to be detectable through preoperative modalities. Considering reports that the prognosis after radical resection for early HCC (including BCLC Stage 0) is favorable and scoring systems indicating that a tumor diameter of ≥2 cm is a prognostic risk factor, performing curative resection for early-stage HCC can be considered one of the treatment options. Therefore, if resection is possible, it can be used as one of the treatment options. (We have added this information to the Discussion p.20, lines 344–351 and References #29,30 in green text). Regarding multiple lesions, we believe that achieving curative resection is more desirable than compromising curability resulting from the inability to resect these small lesions. We believe long-term outcomes are also an important point; however, we have not been able to analyze them thus far. Therefore, this remains a subject for future investigation.

Reviewer#1 point #2: 

What is the cost effectiveness of utilizing an iMRI vs. serial monitoring and f/u with imaging prior to resection once the HCC reaches a larger size? What are the survival outcomes of such as well? The authors discuss tumor free margins but no long-term outcomes.

Author response #2: 

Thank you for your question. Although iMRI-guided hepatectomy is indeed costly, including the investment in equipment, it has the advantage of ensuring patient curability. However, we have not been able to analyze the long-term outcomes thus far; therefore, this remains a subject for future investigation, as mentioned in the limitations section.

Reviewer#1 point #3: 

This study is obviously in comparison to pre-operative ultrasound but what about intra-operative ultrasound? We often confirm all liver lesions with intra-operative ultrasound as it is cheap and widely available at our institution.

Author response #3: 

Thank you for your question. As you mentioned, intraoperative ultrasound is an extremely important tool in hepatectomy for liver tumors, and we routinely use it in all hepatectomies. However, there are small liver tumors that occasionally remain undetected during intraoperative ultrasound examinations. As we mentioned in Table 2 and Discussion (p. 22, lines 382–384), the identification rate of contrast-enhanced intraoperative ultrasonography (CEIOUS) of the target lesions was 36.0%, and 64.0% of them could not be undetectable and unresectable in a usual manner. iMRI may be potentially effective compared with CEIOUS for target lesions in this study; however, further investigation is needed.

Reviewer#1 point #4: 

If the HCCs are <2cm, why were these performed open vs. minimally invasive?

Author response #4: 

Thank you for your question. We believe that laparoscopic surgery is not necessarily essential because the long-term outcomes are comparable between open and laparoscopic surgery. However, we also believe that minimally invasive surgery is preferable, and whether it can be applied to laparoscopic surgery is worth exploring, including the development of new devices. (We have added this information to Discussion, p.22, Lines 390–392 in green text)

Reviewer #2 

Reviewer#2 point #1: 

Please use http://www.ncbi.nlm.nih.gov/mesh

Author response #1: 

Thank you for your suggestion. We have updated the keyword to match the MeSH terminology.

Reviewer#2 point #2: 

Lack of the abbreviations section.

Author response #2: 

Thank you for your pointing this out. We have added a list of abbreviations to the end of the manuscript in red text.

Reviewer#2 point #3: 

Lack of figures?

Author response #3: 

Thank you for your question. It does not appear that there were any missing figures; however, we have uploaded them again.

Reviewer#2 point #4: 

The reference section contains too many old references.

Author response #4: 

Thank you for your pointing this out. We have updated the references by replacing the older ones with newer ones in red text. (Reference #6, #7, #17, #20, and #24)

Reviewer #3 

Reviewer#3 point #1:

The work presented by you is commendable and gives insight into a new modality of treatment for small HCCs of the liver. The manuscript is well written in a logical flow. The results have been written in an organized way discussion is quite relevant. Conclusion is well written.

Author response #1:

Thank you for your positive feedback. We have made revisions based on the comments from other reviewers, and we would appreciate it if you could review the revised sections.

---

## [Decision Letter · Decision Letter 1]

17 Jun 2024

Feasibility, safety, and efficacy of intraoperative magnetic resonance imaging-guided hepatectomy for small hepatocellular carcinoma: A retrospective study

PONE-D-24-13064R1

Dear Dr. Kuroda,

We’re pleased to inform you that your manuscript has been judged scientifically suitable for publication and will be formally accepted for publication once it meets all outstanding technical requirements.

Kind regards,

Takehiko Hanaki, MD, PhD

Academic Editor

PLOS ONE

Additional Editor Comments (optional):

Reviewers' comments:

Reviewer's Responses to Questions

**Comments to the Author**

1. If the authors have adequately addressed your comments raised in a previous round of review and you feel that this manuscript is now acceptable for publication, you may indicate that here to bypass the “Comments to the Author” section, enter your conflict of interest statement in the “Confidential to Editor” section, and submit your "Accept" recommendation.

Reviewer #2: All comments have been addressed

2. Is the manuscript technically sound, and do the data support the conclusions?

Reviewer #2: Yes

3. Has the statistical analysis been performed appropriately and rigorously? 

Reviewer #2: Yes

4. Have the authors made all data underlying the findings in their manuscript fully available?

Reviewer #2: Yes

5. Is the manuscript presented in an intelligible fashion and written in standard English?

Reviewer #2: Yes

6. Review Comments to the Author

Reviewer #2: well-done. congratulations

7. PLOS authors have the option to publish the peer review history of their article (what does this mean?). If published, this will include your full peer review and any attached files.

Reviewer #2: **Yes: **hazim alhiti

---

## [Editor Report · Acceptance letter]

19 Jun 2024

PONE-D-24-13064R1 

PLOS ONE

Dear Dr. Kuroda, 

I'm pleased to inform you that your manuscript has been deemed suitable for publication in PLOS ONE. Congratulations! Your manuscript is now being handed over to our production team.

Kind regards, 

on behalf of

Dr. Takehiko Hanaki 

Academic Editor

PLOS ONE